# Redox-dependent substrate-cofactor interactions in the Michaelis-complex of a flavin-dependent oxidoreductase

Tobias Werther[1], Stefan Wahlefeld[2], Johannes Salewski[2], Uwe Kuhlmann[2], Ingo Zebger[2], Peter Hildebrandt[2] & Holger Dobbek[1]

How an enzyme activates its substrate for turnover is fundamental for catalysis but incompletely understood on a structural level. With redox enzymes one typically analyses structures of enzyme–substrate complexes in the unreactive oxidation state of the cofactor, assuming that the interaction between enzyme and substrate is independent of the cofactors oxidation state. Here, we investigate the Michaelis complex of the flavoenzyme xenobiotic reductase A with the reactive reduced cofactor bound to its substrates by X-ray crystallography and resonance Raman spectroscopy and compare it to the non-reactive oxidized Michaelis complex mimics. We find that substrates bind in different orientations to the oxidized and reduced flavin, in both cases flattening its structure. But only authentic Michaelis complexes display an unexpected rich vibrational band pattern uncovering a strong donor–acceptor complex between reduced flavin and substrate. This interaction likely activates the catalytic ground state of the reduced flavin, accelerating the reaction within a compressed cofactor–substrate complex.

[1] Institut für Biologie, Strukturbiologie/Biochemie, Humboldt-Universität zu Berlin, Berlin D-10115, Germany. [2] Technische Universität Berlin, Institut für Chemie, Sekr. PC14, Straße des 17. Juni 135, Berlin D-10623, Germany. Correspondence and requests for materials should be addressed to P.H. (email: hildebrandt@chem.tu-berlin.de) or to H.D. (email: Holger.Dobbek@biologie.hu-berlin.de).

Enzyme-catalysed reactions require the specific interaction between the enzyme and its substrate in the Michaelis complex. Due to turnover, Michaelis complexes are typically transient making them hard to trap and investigate. This especially hampers crystallographic experiments since the time needed to prepare the Michaelis complex, achieved by soaking crystals with the substrate is usually longer than its lifetime[1]. Thus, to structurally investigate Michaelis complexes, the native substrate is often replaced by a less reactive analogue or competitive inhibitor, or an inactive state of the enzyme is used[1,2]. The latter approach is particularly straightforward for cofactor-dependent enzymes catalysing redox reactions, because they are typically active in only one specific oxidation state. In the case of flavin-dependent oxidoreductases, the oxidized state of the cofactor, which is unable to provide electrons to reduce the substrate, is used to produce non-reactive and thus stable enzyme–substrate complexes mimicking the Michaelis complex (Michaelis mimic). However, enzyme–substrate complexes of several flavoenzymes trapped in this way show a binding of the substrate relative to the cofactor that inhibits hydride transfer from the reduced flavin, thereby impairing the catalytic reaction[3–5]. Thus, to reveal the extent with which substrate–cofactor interactions may depend on the cofactors oxidation state, a direct comparison of substrate–cofactor complexes in the reactive and unreactive state is required.

In this work, we freeze-trapped substrate complexes of a flavoenzyme in its unreactive oxidized and its reactive reduced state (Michaelis complexes) state bound to various substrates. We have employed xenobiotic reductase A (XenA) from *Pseudomonas putida* 86, a versatile catalyst belonging to the old yellow enzyme (OYE) family of flavoproteins. Enzymes of this family catalyse the reduction of the double bond of a wide range of α,β-unsaturated carbonyl compounds, making them attractive for biotechnological applications. XenA catalyses the reduction of 8-hydroxycoumarin in a pathway dedicated to quinoline degradation[3]. The reaction starts with the reductive half-reaction, where the flavin mononucleotide (FMN) cofactor of XenA oxidizes NAD(P)H[6] (Fig. 1). The α,β-double bond of 8-hydroxycoumarin, coumarin as well as the OYE-prototypical substrate 2-cyclohexenone are reduced by XenA in the following oxidative half-reaction (Fig. 1), which is supposed to proceed by a *trans*-addition of a hydride from flavin N5 and a proton from Tyr183 (ref. 7). Here, we employed a variant of XenA in which the proton-donating Tyr183 is exchanged against phenylalanine (Y183F-XenA), selectively slowing down the oxidative half-reaction[7]. Using Y183F-XenA, we prepared stable, yet minimally perturbed Michaelis complexes, which we investigated using X-ray crystallography, stopped-flow kinetics and resonance Raman (RR) spectroscopy. The combined study provides direct evidence for oxidation-state-dependent substrate–cofactor interactions and shows that cofactor compression creates a strong donor–acceptor complex between reduced flavin and the substrates.

## Results

**Stopped-flow spectroscopic analysis.** We showed previously that the Y183F-substitution Y183 slows down the reoxidation of reduced XenA with 2-cyclohexenone by two orders of magnitude[7]. To evaluate if this would also allow us to stabilize and freeze-trap XenA-substrate complexes, we recorded time-dependent spectral changes during the reoxidation of wild-type (wt) and Y183F-XenA with coumarin substrates (coumarin and hydroxylated derivatives). We observed that reaction intermediates with distinct substrate-specific ultraviolet/visible spectral features are formed within the dead-time of the experiment (Fig. 2), resembling the respective Michaelis complexes.

The most characteristic spectrum was found for the Michaelis complex with coumarin, which displays a peak at 480 nm (wt) or 506 nm (Y183F), respectively (Fig. 2c). Michaelis complexes with 7-hydroxycoumarin and 8-hydroxycoumarin show only a shoulder between 450 and 600 nm. The fact that only absorption shoulders and not maxima were observed is likely caused by the strong absorption of the free hydroxycoumarins, which dominates the spectra at wavelength shorter than 450 nm (Fig. 2c). Spectra of the reduced complexes of wt- and Y183F-XenA with coumarin resemble spectra of reaction intermediates of other flavin-dependent enzymes such as D-amino acid or L-phenylalanine oxidases[8–10] and acyl-CoA dehydrogenases[11–13].

Wt-XenA reoxidizes quickly, indicated by an increase of the absorption around 460 nm (Fig. 2). When reduced wt-XenA reacts with a hydroxycoumarin an additional absorption band slowly arises in the long-wavelength region, which originates from a charge-transfer complex between hydroxycoumarin and reoxidized XenA. The same long-wavelength band is observed when oxidized XenA is incubated with a hydroxycoumarin[3]. Generally, Y183F-XenA re-oxidizes with all three coumarin substrates by at least two orders of magnitude slower than wt-XenA and no reoxidation is observed in the presence of coumarin (Supplementary Table 1). As the substrate complexes of reduced Y183F-XenA exhibit life-times of hours, they can be sufficiently populated by soaking reduced Y183F-XenA crystals with coumarins.

**Crystal structures of Michaelis complexes.** We solved crystal structures of reduced as well as oxidized Y183F-XenA with and without the three coumarin substrates. The resulting eight crystal structures were determined at resolutions of $d_{min} = 1.44$–$1.07$ Å (Supplementary Table 2). Michaelis complexes were freeze-trapped after soaking crystals of reduced Y183F-XenA with substrates under anoxic conditions. Substrate molecules that accumulated in the active sites turned the colourless-reduced crystals into the same colour as observed for these complexes in solution (Supplementary Fig. 1).

Crystal structures of the coumarins with reduced and oxidized Y183F-XenA differ remarkably. All three coumarins bind coplanar on top of the isoalloxazine ring, but their orientations depend on the oxidation state of the flavin. 8-Hydroxycoumarin binds to reduced Y183F-XenA (Fig. 3a) in an inverse orientation compared to the non-productive complex found with oxidized Y183F-XenA (Fig. 3b). All coumarins are hydrogen-bonded to the histidine pair, His178 and His181 of Y183F-XenA. In the oxidized complex, the histidine pair binds the hydroxyl group of 8-hydroxycoumarin, whereas in the reduced complex it interacts with the carbonyl oxygen of the substrate. Thereby, in the reduced complex the distance between the hydride donor and acceptor is shortened and thus becomes catalytically competent (C4$_{8HC}$—N5$_{FMNred}$: 3.14 Å versus C4$_{8HC}$—N5$_{FMNox}$: 4.23 Å; Fig. 3a)[14].

We observed a similar situation for the reduced complex containing 7-hydroxycoumarin (Fig. 3c). 7-Hydroxycoumarin is pseudo-symmetrical relative to the C4A/C1A axis (annulated carbon atoms), making its orientation in the electron density ambiguous (Supplementary Fig. 2). Therefore, we refined 7-hydroxycoumarin in the two possible substrate orientations (in both oxidation states of Y183F-XenA) and deduced the correct substrate orientations from the B-factor distributions (Supplementary Fig. 2). As seen for 8-hydroxycoumarin, the histidine pair 178/181 interacts with the hydroxyl group of 7-hydroxycoumarin in the oxidized and with the carbonyl oxygen in the reduced complex, again decreasing the distance between reacting atoms in the reduced compared to the oxidized complex (C4$_{7HC}$—N5$_{FMNred}$: 3.14 Å versus

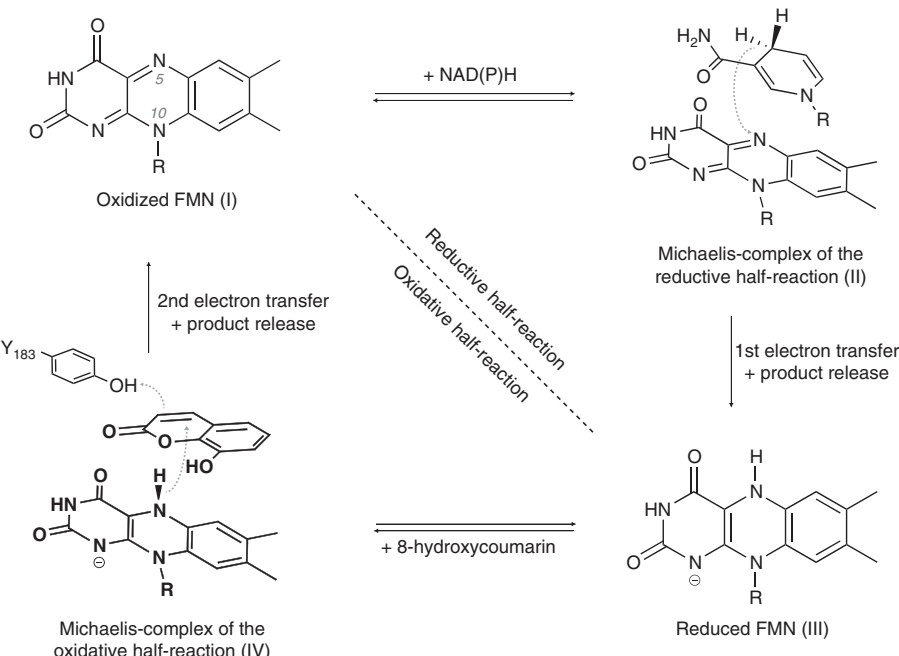

**Figure 1 | Reaction mechanism of the flavin-dependent oxidoreductase XenA.** XenA catalyses the NAD(P)H-dependent reduction of various α,β-unsaturated carbonyl compounds like 8-hydroxycoumarin following a ping-pong mechanism[3,6]. The overall reaction cycle can be divided into a reductive and an oxidative half-reaction, reflecting the oxidation state changes of the enzyme-bound flavin mononucleotide cofactor (FMN). Both half-reactions proceed via Michaelis complexes (II, IV). The proposed mechanism of the oxidative half-reaction proceeds by a *trans*-addition of a hydride transfer from flavin N5 and a proton from Y183 acting as a general acid by protonating the α-carbon of the α,β-carbon double bond of the substrate[7,43,52].

$C4_{7HC}$—$N5_{FMNox}$: 3.95 Å). Thus, an efficient hydride transfer from the flavin to the substrate can only occur in the orientation found in the reduced complex, where the reaction centres are suitably aligned[14].

Coumarin is a substrate of XenA that lacks a hydroxyl group. Its binding modes within both the non-productive complex with oxidized Y183F-XenA and the reactive Michaelis complex with reduced Y183F-XenA are identical (Fig. 3d). In both oxidation states, the carbonyl oxygen of coumarin interacts with the histidine pair and reactive atoms are close to each other ($C4_{HC}$—$N5_{FMNred}$: 3.21 Å versus $C4_{HC}$—$N5_{FMNox}$: 3.06 Å). Thus, we conclude that the phenolic hydroxyl group of hydroxycoumarins is crucial for the observed redox-dependent binding.

The true-atomic resolution of our complex structures provides information beyond the substrate-binding mode and allows to assess and compare the precise geometry of bound substrates. Bound 8-hydroxycoumarin has a significantly shorter C8–O8 bond length in the oxidized than in the reduced complex structure (1.28 versus 1.39 Å) (Supplementary Fig. 3a). Furthermore, the oxidized and reduced structure differs in the C1A–C8–O8 angle of 8-hydroxycoumarin (Supplementary Fig. 3a). These geometry changes can be explained by different protonation states of 8-hydroxycoumarin. Structure optimizations of protonated and deprotonated 8-hydroxycoumarin using density functional theory (DFT) (B3LYP/6-31Gd), reproduce the geometry changes observed in the crystal structures and indicate that the C8–O8 bond length and C1A–C8–O8 angle are particularly affected by the protonation state (Supplementary Fig. 3a). We therefore investigated by proton linkage experiments if the hydroxycoumarins deprotonate upon substrate binding (Supplementary Fig. 3b). When the two hydroxycoumarins bind to oxidized XenA, a proton release to the buffer is observed, while binding of the coumarin lacking a hydroxyl group results in no significant net proton uptake or release (Supplementary Fig. 3b).

Thus, the hydroxycoumarins are stabilized in the non-productive oxidized complex in their deprotonated phenolate state. Binding of an anionic phenolate to an oxidized flavin has also been observed in related enzymes belonging to the OYE family, resulting in charge-transfer complexes[15–17] with spectra similar to the ones we observed for the complexes of oxidized XenA with 7-hydroxycoumarin and 8-hydroxycoumarin (Fig. 2a,b and Supplementary Fig. 1). In consequence, the flavin determines by its oxidation state if the substrate binds in the protonated or deprotonated state, which in turn determines how the substrate aligns itself in the active site.

When the FMN in XenA becomes reduced, the dihedral angle between the pyrimidine and dimethylbenzene rings, the so-called flavin butterfly bent angle, increases from 15.0° (oxidized FMN) to 21.9° (reduced FMNH$^-$) (Supplementary Fig. 4). In addition, the central pyrazine moiety of the isoalloxazine ring switches upon reduction into a boat conformation, in which the N5 and N10 atoms point towards the *si*-face of the isoalloxazine ring (Supplementary Fig. 4). Similar conformational changes of the isoalloxazine ring upon reduction have been observed for other flavoproteins[5,18–21]. Interestingly, when the coumarin substrates bind to Y183F-XenA, they alter the conformation of the reduced flavin (Supplementary Fig. 4). 8-Hydroxycoumarin binds almost coplanar on the *si*-face of the isoalloxazine ring, closer than expected for van der Waals interactions between cofactor and substrate. The binding of 8-hydroxycoumarin reduces the flavin butterfly bend angle, from 21.9° to 14.3°. Moreover, 8-hydroxycoumarin pushes the N5 atom 0.23 Å downwards to the *re*-face of the cofactor in the Michaelis complex. The same effect is observed for the Michaelis complex of reduced flavin with coumarin and 7-hydroxycoumarin (Supplementary Fig. 4). Thus, upon binding the substrates force the reduced cofactor into a conformation more similar to that of the oxidized state (Supplementary Fig. 4).

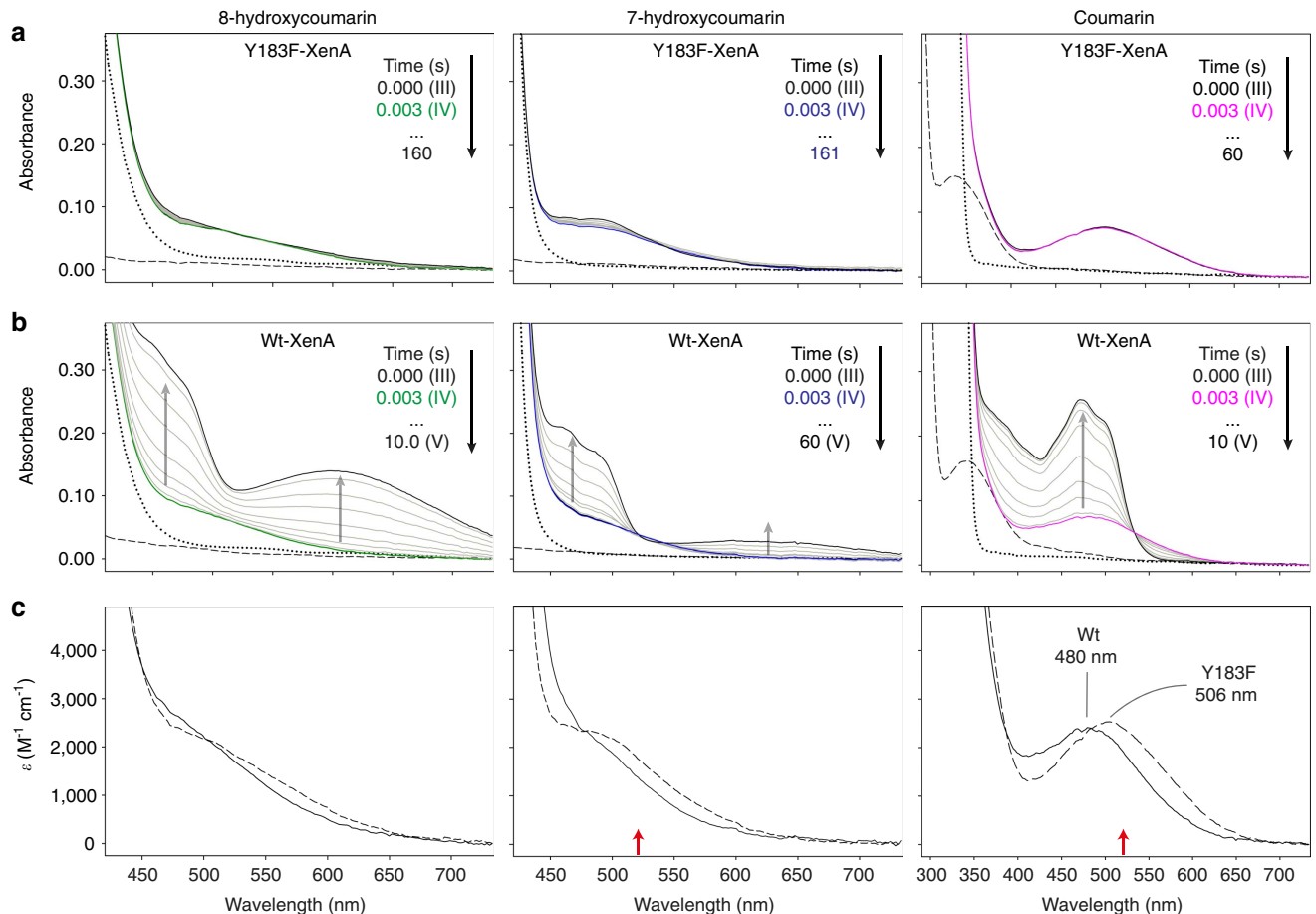

**Figure 2 | Kinetics of the reoxidation of XenA.** Time-dependent spectral changes for the reoxidation of 30 µM reduced (**a**) Y183F-XenA and (**b**) wild-type XenA by 8-hydroxycoumarin, 7-hydroxycoumarin and coumarin. The dashed and dotted lines represent the spectra of the reduced enzymes and substrate, respectively. Grey arrows indicate the direction of absorbance changes. The first spectra after 3 ms represent the Michaelis complexes (IV) of the oxidative half-reaction with 8-hydroxycoumarin (green), 7-hydroxycoumarin (blue) and coumarin (pink), respectively. The black bold spectra, obtained after complete reoxidation of reduced wt-XenA, correspond to the non-productive Michaelis complex mimics (V). (**c**) Spectral overlay of the reaction intermediates (Michaelis complexes; IV) of wt-XenA (solid-line) and Y183F-XenA (dashed line). The bathochromic shift in the Y183F-variant may reflect minor changes in the microenvironment due to the mutation. The red arrows indicate the excitation wavelength used in RR experiments.

**Resonance Raman spectroscopy of XenA complexes**. To further probe the substrate–cofactor interaction, we used RR spectroscopy and compared the oxidized Michaelis complex mimics with the reduced Michaelis complexes. RR spectra were measured from oxidized and reduced Y183F-XenA solutions in presence and absence of the substrates coumarin or 7-hydroxycoumarin, respectively, using 514 nm excitation. RR spectra of the oxidized XenA obtained in this way show a rich vibrational band pattern with well-resolved bands, characteristic of the oxidized flavin (Supplementary Fig. 5). The bands can readily be assigned on the basis of DFT calculations of the flavin cofactor *in vacuo* (Supplementary Table 3) and are in good agreement with previous empirical normal mode analyses[22–24]. The oxidized complexes formed with coumarin and 7-hydroxycoumarin reveal very similar RR spectra, except for small band shifts that do not exceed 4 cm$^{-1}$ (Supplementary Fig. 5). These shifts predominantly refer to vibrational modes localized in the pyrimidine and pyrazine moieties (rings II and III) of the flavin, above which the substrates are coplanarily bound as shown by the crystal structures (Fig. 3 and Supplementary Fig. 7). Thus, the observed spectral changes may reflect the sterical compression of the oxidized isoalloxazine ring system upon substrate binding, as indicated by the reduction of both the flavin butterfly bend angle and the atomic displacement factors

(Supplementary Fig. 7 and Supplementary Fig. 4). However, the spectra do not indicate any significant electronic interactions between the cofactor and the substrate in the ground state (*vide infra*).

Note that the species of the 7-hydroxycoumarin complex giving rise to the broad CT transition at ca. 600 nm (Supplementary Fig. 1) does not contribute to the RR spectrum measured at 514 nm excitation.

Reduced XenA displays a different picture. Whereas, the substrate-free reduced flavin does not show any RR bands due to the lack of resonance enhancement at 514 nm, complex formation with both coumarin and 7-hydroxycoumarin affords RR spectra with an unexpectedly large number of bands in the region between 1,000 and 1,700 cm$^{-1}$ (Supplementary Fig. 5). The spectra differ substantially from those of the oxidized flavin and the reduced flavin reported in the literature[25] but also bear no resemblance to those of CT complexes obtained from substrate binding to reduced flavoproteins[8–11,26]. Spectra of such complexes are comparatively featureless showing only one characteristic band at ca. 1,610 cm$^{-1}$, attributable to the flavin. All other peaks in the RR spectra of these CT complexes have been assigned to bound products or substrate analogues[9,12,27]. Instead, the present RR spectra of the reduced substrate-XenA complex are not the sum of FMNH$^-$ and substrate spectra. This is

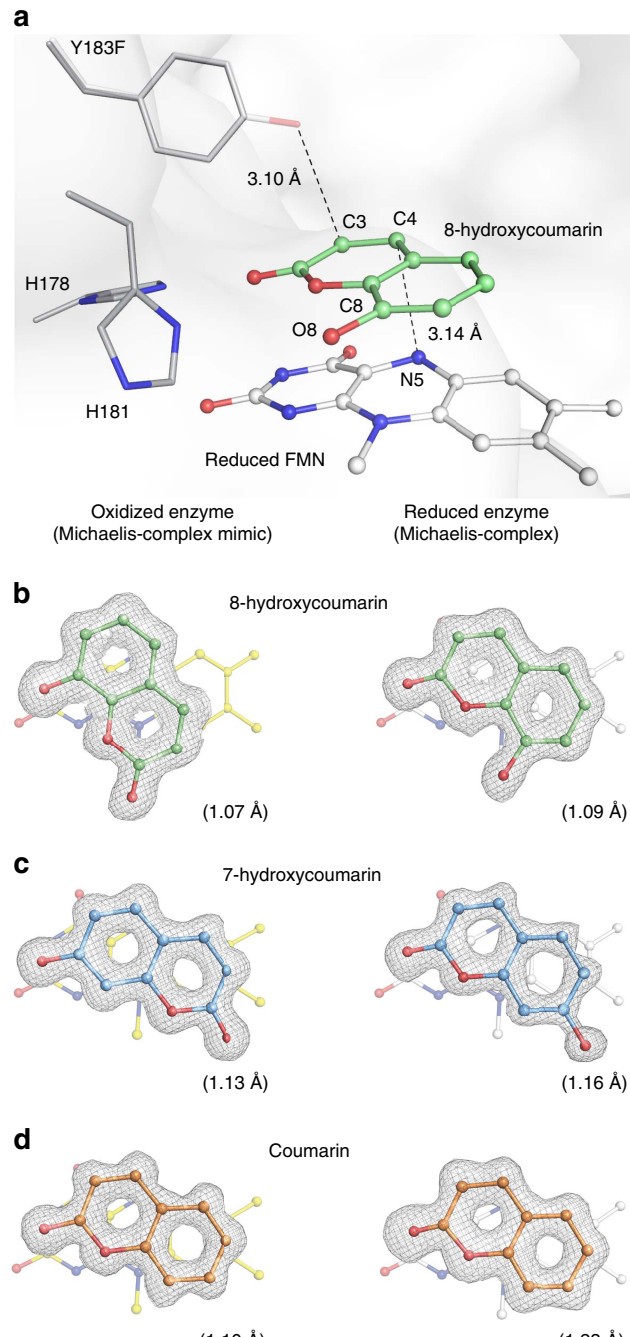

**Figure 3 | Oxidation-state-dependent substrate-binding to the active site of XenA.** (**a**) Active site view of the Michaelis complex between reduced Y183F-XenA and 8-hydroxycoumarin (green sticks). The Y183 hydroxyl group of wt-XenA is depicted semitransparent. The dashed lines illustrate the distances between the α,β-carbon double bond of the substrate with either the N5 atom of the isoalloxazine ring or the oxygen atom of the Y183F hydroxyl group. (**b,c**) View on the *si*-face of FMN illustrates the different orientations that 8-hydroxycoumarin (**b**) and 7-hydroxycoumarin (**c**) adopt in the oxidized and reduced enzyme. (**d**) The hydroxyl group lacking substrate coumarin (orange carbons) binds in the same orientation regardless of the oxidation state of the cofactor. The *Fo−Fc* omit maps of the substrate molecules are contoured at 4.0 σ. The values in parenthesis give the resolution of the diffraction data used for refinement ($d_{min}$).

illustrated by a closer inspection of the region between 1,500 and 1,700 cm$^{-1}$ for the reduced XenA-coumarin complex (Fig. 4). The band fitting analysis reveals 12 bands in this region which is nearly two times larger than the sum of the Raman-active modes of the individual components predicted by DFT calculations. Even including the (two) Raman-inactive modes, the number of experimental bands is still distinctly larger. This is also true considering the theoretical number of normal modes for other cofactor–substrate combinations such as an anionic flavin semiquinone and a neutral coumarin radical. Furthermore, most of the bands display unprecedented small band widths of ca. 6 or 8 cm$^{-1}$ that are nearly two times smaller than typically observed in RR spectra of cofactors in proteins[28] including the oxidized XenA complexes studied in this work. Finally, one may identify a certain pattern of spacing between the individual bands, most clearly visible for the band centred at 1,562.2 cm$^{-1}$ with two satellite bands ca. 6–7 cm$^{-1}$ on the high- and low-energy side. Similar satellite bands with a spacing of ca. 9 cm$^{-1}$ may be associated with the 1,592.2 cm$^{-1}$ band. Here the higher frequency side band at 1,604.5 cm$^{-1}$ seems to overlap with the lower frequency component of the triplet centred at 1,613.4 cm$^{-1}$. In a similar way, one may interpret also the lower frequency region between 1,000 and 1,500 cm$^{-1}$ although the distinctly higher density of modes aggravates disentangling main band components from satellite bands (Supplementary Fig. 5). The spectra measured from samples in D$_2$O and of the 7-hydroxycoumarin complex display a qualitatively similar behaviour (Supplementary Fig. 6 and Supplementary Table 4).

Splitting of vibrational modes is known for identical molecules in crystals (Davydov or factor group splitting)[29,30] or two-dimensional crystalline-like packing in monolayers[31], which cannot account for the present observations in the RR spectrum of a molecular complex composed of two different molecules of low symmetry. Instead, a possible explanation roots in the fact that upon complex formation six low-frequency intermolecular vibrational modes are formed that are derived from three translational and three rotational degrees of freedom[32]. Some of

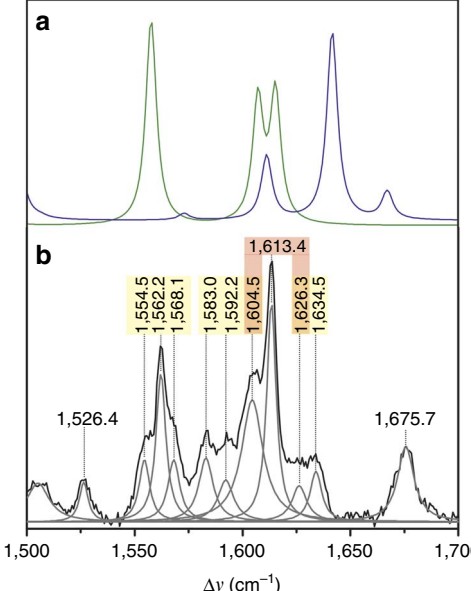

**Figure 4 | Raman spectra of XenA.** (**a**) Calculated spectra of reduced flavin (blue trace) and coumarin (green trace); (**b**), experimental RR spectrum of the reduced complex of Y183F-XenA and coumarin measured with 514-nm excitation. Yellow- and red-shaded peak labels indicate conjugate bands (triplets, doublets).

these modes may combine additively and substractively with intramolecular fundamentals of either the flavin or the coumarin component, leading to combination modes which together with the fundamental itself may form a triplet. The intrinsic intensity of combination modes is rather low but they may be enhanced via Fermi resonance as observed for complexes of organic molecules[33]. Note that earlier studies on the coupling of intra- and intermolecular modes referred to weak molecular complexes with less severe perturbations of the spectra of the components than observed in the present case[32,33]. Thus, we conclude that substrate binding to reduced XenA leads to strong molecular complexes between the reduced flavin and the coumarin, associated with a charge transfer in the ground state (ground state CT complex). Unfortunately, an adequate theoretical treatment of such a strong donor–acceptor complex inside the protein matrix was beyond our capacities.

## Discussion

Using the catalytically impaired Y183F-XenA variant, we stabilized the reduced Michaelis complex of a flavin-dependent oxidoreductase and thus were able to resolve its structural and electronic features. The observed properties and drawn conclusions most likely apply also to wt-XenA, as the ultraviolet/visible spectral characteristics of the stabilized Y183F-coumarin complexes are very similar to the respective kinetically competent intermediates observed for the complex 3 ms after mixing reduced wt-XenA with coumarins (Fig. 2 and Supplementary Fig. 1); the crystal structures of the oxidized Michaelis complex mimics for both wt-XenA and Y183F-XenA are practically identical, except for the missing hydroxyl group, and the dissociation constants of the reduced Michaelis complexes for the respective substrates are in the same range for wt-XenA and Y183F-XenA, indicating that the amino acid exchange barely affects the complex (Supplementary Table 1). Consequently, the crystal structure and vibrational spectra observed for the complex between substrate and reduced flavin in Y183F-XenA closely resemble that of a substrate enzyme complex at the onset of turnover—an authentic Michaelis complex. Typically, amino acid side chains in the active site determine where and how a substrate binds. In XenA, while most side chains are preoriented for substrate binding, Trp302 can change its conformation in response to substrate binding[34] or the flavin oxidation state[7] (Supplementary Fig. 8). This work suggests that the substrate-binding mode can also be altered by the oxidation state of the flavin cofactor.

We argue that the electronic structure of the isoalloxazine ring, which changes upon flavin reduction, modulates the $pK_a$ of ionizable groups in bound substrates. In this way, the oxidized, electron-deficient flavin[35] stabilizes the phenolate anion, making the anionic hydroxycoumarin a CT-donor for oxidized XenA, while the reduced, electron-rich flavin favours binding of the protonated substrate, whose enone function makes it a potential CT-acceptor for reduced XenA. While the protonation state of the substrate is modulated by the flavin oxidation state, it is the interaction of the His pair with the phenolate oxygen (oxidized XenA) compared to the carbonyl oxygen (reduced XenA) that determines the orientation of the bound hydroxycoumarins. Oxidation-state-specific electrostatic interactions between the cofactor and the substrate can thus generate an additional level of control, perhaps reducing the number of futile enzyme–substrate encounters with unproductive substrate orientations[36].

This additional control has consequences for the interpretation of crystal structures: Structures of Michaelis complex mimics do not necessarily reflect the structures of the Michaelis complex and have to be treated with care before mechanistic conclusions

are drawn. Furthermore, oxidation-state-dependent cofactor electrostatics need to be considered when adapting flavin oxidoreductases for new substrates in biotechnological applications and to increase the yield of optically pure building blocks—an area especially active for XenA and homologues[37–39].

We reported earlier that coumarin binds coplanar on top of the oxidized isoalloxazine ring and compresses it to a more planar conformation than observed without ligand[40]. At that time, we only investigated the unreactive Michaelis mimic. In this work, we confirmed the tight binding in the Michaelis mimic, but more importantly, found that active site compression resulting in a more planar isoalloxazine ring is even more pronounced in the reduced Michaelis complex (Supplementary Fig. 4), where it is relevant for substrate turnover.

The RR spectra complement the structures as they show that binding of coumarins to oxidized (Michaelis mimic) and reduced XenA (Michaelis complex) results in different types of interactions: a coumarin molecule bound on top of the oxidized isoalloxazine ring merely shifts the vibrational bands of the cofactor by a few wavenumbers, but when it binds to the reduced isoalloxazine ring, substrate and cofactor create a strong donor–acceptor complex with properties unlike that of the individual components. How much electronic charge is transferred in this complex from the donor FMNH⁻ to the charge-accepting coumarins is unknown, but the number and splitting of the RR bands suggests a strong interaction with substantial charge flow in the ground state of the complex.

Thus, the Michaelis complex of XenA is characterized by active site compression, increased planarity with reduced N5-pyramidalization of the flavin and a strong donor–acceptor interaction between cofactor and substrate – three likely interdependent properties. Whereas the cofactor becomes strained by substrate binding, the coumarin substrates are not visibly distorted but may be stressed[41], as indicated by short vdW-contacts. When a reduced flavin is forced to adopt a more planar structure, it becomes destabilized, gets structurally more similar to the oxidized state and is transformed to a more potent reductant[42]. Thereby, the steric compression of the reduced flavin in the Michaelis complex may activate the ground state by destabilizing the cofactor and engaging it in strong electronic interactions. As this primarily depends on the flavin and its substrate, it may be a general strategy of flavoenzymes.

## Methods

**Protein expression and purification.** Mutagenesis to derive the Y183F-XenA variant has previously been described[7]. Both mutant and wild-type genes were subcloned into pET28a vector (Novagen), containing a N-terminal His-tag and an additional tobacco etch virus (TEV)-protease cleavage site. The His-tagged protein was expressed in *Escherichia coli* Rosetta (DE3) using Terrific broth media supplemented with 50 μg ml⁻¹ kanamycin and 34 μg ml⁻¹ chloramphenicol. Cells were grown at 37 °C until OD₆₀₀ reached 0.6. After addition of 100 μM IPTG the temperature was decreased to 17 °C and the genes were expressed overnight. After cell lysis by sonication, XenA was purified by nickel affinity chromatography. The pooled fractions were treated with TEV-protease overnight at 8 °C. The His-tag, residual uncut protein as well as the TEV-protease were removed by a second nickel affinity purification. Cutted protein was concentrated, reconstituted with 5 mM FMN, and loaded onto a size-exclusion chromatography column to remove excessive cofactor. Finally, the concentrated protein was frozen in liquid nitrogen and stored at −80 °C.

**Pre-steady state kinetics and ITC experiments.** All experiments were performed in 50 mM Tris/HCl buffer pH 8.0 at 25 °C. The oxidative half-reactions of both reduced wt-XenA and the Y183F-XenA variant (5 μM for photomultiplier tube detection or 30 μM for diode array detection) with 8-hydroxycoumarin, 7-hydroxycoumarin and coumarin were initiated with a Biologic SFM 400 stopped-flow spectrophotometer and recorded either with a diode array detector (J&M TIDAS) or photomultiplier tube detector (R374, Hamamatsu Phototonics), using a cuvette with 1-cm optical pathlength. To prevent oxidase activity, the measurements were performed under strict anoxic conditions (< 0.5 p.p.m. O₂) within a glove box (MBraun, Garching, Germany). Complete reduction of the

enzyme was achieved by incubation with 2 mM glucose-6-phosphate, 2.5 U ml$^{-1}$ glucose-6-phosphate dehydrogenase (Calbiochem), and 0.3 μM NAD$^+$ for 2 h (ref. 43). The measurements were performed in triplicates.

Isothermal titration calorimetry (ITC) experiments (VP-ITC Microcalorimeter, GE Healthcare) were performed to determine the thermodynamic constants of binding for the substrates 8-hydroxycoumarin, 7-hydroxycoumarin and coumarin to reduced Y183F-XenA. To measure the effect of proton linkage upon ligand binding[44], ITC experiments were carried out at pH 7.5 in 20 mM buffers with distinct deprotonation enthalpies ($\Delta H_{ion}$)—Hepes/NaOH (5.02 kcal mol$^{-1}$), Tricine/NaOH (7.64 kcal mol$^{-1}$), Tris/HCl (11.35 kcal mol$^{-1}$), and phosphate buffer (1.22 kcal mol$^{-1}$)[44]. The ionic strength was kept constant at 100 mM by adding NaCl. Thermograms were integrated with NITPIC[45] and the resulting isotherms were analysed with the ORIGIN 2.7 software.

**Crystallization and structure determination.** Crystals of reduced Y183F-XenA were grown under anoxic conditions in a glove box (Model B, Coy Laboratory Products, Michigan) using the same crystallization condition as described for wt-XenA[3], additionally containing 4 mM NADH and catalytic amounts of catalase (Sigma, Aldrich). Crystals were incubated for 5–10 min in reservoir buffer supplemented with the oxidizing substrate (12 mM 8-hydroxycoumarin, 3 mM 7-hydroxycoumarin, 4 mM coumarin) and 25% (v/v) glycerol, and were flash-frozen in liquid nitrogen. X-ray diffraction data were collected at the Helmholtz Zentrum Berlin beamlines BL14.1 and BL14.2 (Berlin, Germany), at a wavelength of 0.91841 Å and 100 K (ref. 46). Diffraction data were processed and scaled using the XDS package[47]. The Y183F-XenA structures were solved by molecular replacement using the program PHASER[48] and oxidized wt-XenA (PDB-Id: 3L5L) as search model[40]. The structure was build and refined using the programs COOT[49] and PHENIX[50]. All structures were refined with riding hydrogen atoms. For structures with $d_{min} < 1.5$ Å, B-factors were refined anisotropic for all non-hydrogen atoms except water molecules. The bound substrate molecules and the isoalloxazine ring of FMN were refined with restrains to allow bending and twisting. The flavin butterfly bend angle was calculated as the angle between two intersecting planes using MATLAB 8.0.0. The planes were determined by least-square fitting through either the six atoms of the benzene ring or the pyrimidine ring, plus the N5 and N10 atoms of the flavin. Ligands bound in the active site are well defined in the electron density (Supplementary Fig. 9).

**Resonance Raman spectroscopy.** All samples were prepared inside a glove box (VAC, Hawthorne, CA, USA) to prevent oxidation of reduced complexes. Anoxic buffers were prepared by at least five cycles of evacuating and flushing with nitrogen gas at a vacuum gas line. Substrate solutions were prepared always fresh in anoxic buffers and the concentration was determined spectroscopically ($\varepsilon_{(280nm)} = 10,400$ M$^{-1}$ cm$^{-1}$ for coumarin and $\varepsilon_{(365nm)} = 12,650$ M$^{-1}$ cm$^{-1}$ for 7-hydroxycoumarin). For the RR experiments, samples were prepared by exchanging buffer of concentrated protein stock solutions against 50 mM Tris/HCl pH 8.0, 150 mM NaCl containing the respective substrate using a PD10 column (GE Healthcare). For the reduced complexes, the protein sample was reduced by excess of NADH prior application onto the PD10 column. Afterwards, the samples were concentrated using Vivaspin 500 concentrators (10.000 MWCO, Sartorius Stedim Biotech GmbH) and were either directly placed on a quartz holder or frozen in liquid nitrogen until further use. Samples were cooled to 80 K using a liquid nitrogen cooled cryostat (Linkam). Samples were excited at 514 nm using an argon laser (Innova 70, Coherent). The typical power at the sample was 1 mW. The spectrometer was calibrated before each experiment with toluene as external standard. After background subtraction, some of the spectra were analysed by a band fitting procedure using Lorentzian functions[28].

**Quantum chemical calculations.** For the FMN, coumarin, and 7-hydroxy-coumarin molecules geometry optimization and calculation of Raman spectra were performed using the software package Gaussian09 (ref. 51) on the BP86 level of theory with a 6–31G* basis set applied on all atoms. Convergence criteria were set to 'tight'. Calculations were performed for the fully oxidized, fully reduced and semiquinone radical flavin structure, the latter being assumed as either deprotonated (anionic), neutral, or protonated (cationic). Coumarin and 7-hydroxycoumarin were considered as neutral molecule or reduced species carrying either a negative charge (anionic radical) or a hydroxyl group (neutral radical). The solvent-exposed sites for FMN and substrate binding were mimicked by employing a polarizable continuum model using the relative permittivity of water $\varepsilon_r(H_2O) = 78.36$, as implemented in Gaussian09.

**Data availability.** Refined protein structures and corresponding structure factor amplitudes have been deposited in the PDB under accession codes 4uth, 4uti, 4utj, 4utk, 4utl, 4utm, 5lni and 5lnj. Data supporting the findings of this study are available within the article (and its supplementary information files) and from the corresponding authors upon reasonable request.

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

## Acknowledgements

The Deutsche Forschungsgemeinschaft is acknowledged for funding of the project (EXC 314 'Unifying Concepts in Catalysis—UniCat'). We acknowledge access to beamlines of the BESSY II storage ring (Berlin, Germany) via the Joint Berlin MX-Laboratory sponsored by the Helmholtz Zentrum Berlin für Materialien und Energie, the Freie Universität Berlin, the Humboldt-Universität zu Berlin, the Max-Delbrück-Centrum and the Leibniz-Institut für Molekulare Pharmakologie. We thank Olivia Spiegelhauer-Hartl for providing Y183F-XenA protein for initial studies.

## Author contributions

T.W. designed research, prepared protein samples, recorded and analysed crystallographic, kinetic and binding data, analysed RR data, and wrote the manuscript. S.W. recorded and analysed RR spectra, and prepared RR related figures. J.S. calculated spectra of isolated flavin and substrate species. U.K. and I.Z. analysed RR spectra. P.H. and H.D. designed research, analysed data and wrote the manuscript.

## Additional information

**Competing interests:** The authors declare no competing financial interests.

