## [Peer Review File · Nature Communications]

Reviewer #1 (Remarks to the Author):

In this carefully executed study, Werther et al investigates the structural and spectroscopic properties of a NADPH-dependent flavoenzyme reductase bound to the substrate in both oxidized and reduced states. By making use of a mutant, the authors were able to trap in the enzyme in a Michaelis complex which outlines the structural and geometric feature of the flavin-catalyzed reduction step. The strong points of this work are manifold: (i) the high resolution of the Xray analysis, (ii) the clever usage of a active-site mutant, and (iii) the use of Resonance Raman spectroscopy to characterize the enzyme (possibly the most innovative aspect of this work). On the other hand, there a few weak points that cannot pass unnoticed. The initial claim that there are no structures of authentic Michaelis complexes between reduced flavoenzymes and substrates is exaggerated. In the literature there as several examples of structures obtained by soaking flavoenzyme crystals in substrates, leading to the formation of reduced enzyme-product complexes (exactly the same state reported in this work, just in the opposite reaction direction). Classic examples are e.g. D-amino acid oxidase and cholesterol oxidase (reported at super high resolution). Indeed, the structures reported in this work confirm the previous observations on the structural bases of flavoenzyme mechanisms with the added values of the electronic properties inferred from the Raman spectra. Likewise, it has been often found that flavins can slightly deviate from planarity depending on the redox and ligation state. Similarly, the protonation state of the ligands have been reported before to be affected by the flavin being reduced or oxidized. In this context, another limitation from this study is that no spectra (e.g. absorbance and/or Raman) were measured on the crystals to allow a better correlation between solution and crystal experiments. Finally, it is somewhat surprising that there is no mention for the reasons making the mutant slower compared to the WT enzyme. In summary, these are very well performed experiments, which will be of interest to (flavo)enzymologists. However, these data seem to be more of incremental/confirmative nature rather than outlining new concepts or mechanisms in enzymology.

Reviewer #2 (Remarks to the Author):

There is so much to like in this manuscript that I feel like a curmudgeon complaining about its important shortcomings. Splendid high-resolution crystal structures are presented of the reduced flavoenzyme XenA thwarted from reacting with coumarin substrates by mutagenesis; absorbance spectra document the charge-transfer interactions between the oxidized and reduced states of the mutant enzyme and the coumarin derivatives; resonance Raman spectra further document the vibrational modes of these complexes; and redox and binding reactions (at least some) are also reported. The problem is that these beautiful data aren't matched by adequate interpretations, leading to big claims that are poorly-supported (or just poorly-argued?).

The sorest point of the manuscript is the resonance Raman spectroscopy. It is implied on page 9 that the complexity of the RR spectra of the CT complexes with reduced enzyme is unexpected, with rather old references being cited for the precedence of simplicity. The technology of RR spectroscopy has improved markedly in the past two to three decades; those older "simple" spectra no doubt illustrate the vast improvements in technology that enabled the beautiful detailed spectra in this manuscript. Indeed, RR spectra of CT complexes to oxidized flavoenzymes [J. Raman Spectrosc. (2001) 32, 579-586] shows plentiful complexity. Why should CT complexes with reduced enzyme be any less complex? Another important misinterpretation is the attempt to explain the supposedly surprising complexity of the RR spectrum by invoking the presence of significant amounts of anionic semiquinone formed by a frank electron-transfer in the CT complexes. This proposal is refuted by the absorbance spectra of the CT complexes, which would be quite sensitive to semiquinone but, in fact, show none. The current proposal seems to revive the debates from a half-century ago over the nature of flavoprotein CT complexes, without offering a compelling analysis that would change, for instance, the tightly-argued conclusions in reference 13 that Muliken's analysis of CT complexes from 1952 applies. A resonance form of the CT complex may indeed resemble a semiquinone-hydroxy coumarin radical pair, and maybe a VB calculation could assess how much such a form contributes, but this should not be misinterpreted as a true radical pair. With such an unorthodox interpretation of the RR spectra, which are admitted to be beyond adequate theoretical capabilities, and the detail devoted to arguing that many observed RR bands are actually composed of several independent overlapping resonances, further support should be sought, especially from the spectra of isotopically labeled flavin and hydroxy coumarins.

This report confirms a proposal previously published by these workers that hydroxy coumarins change their binding orientation by $\sim 180^\circ$ upon flavin reduction from the unreactive configuration previously reported for the dead-end complex with oxidized enzyme to the analog of the reactive form seen in this work with the reduced mutant enzyme. The authors rightly emphasize the broader implication of this result, cautioning that crystal structures of dead-end complexes might not represent the actual reactive complex. This is an important message. That said, it was disappointing that a detailed explanation wasn't given for the phenomenon in this example. It's unassailably vague to say that there was a "change in the microenvironment". The behavior of XenA likely is a collaboration between the curious ambidextrous nature of the hydroxy coumarins, which are electron-rich on the phenolic side, making them CT-donors when bound to the oxidized enzyme but whose enone makes them CT-acceptors when bound to the reduced enzyme; and the pair of histidines close to N1-C2O of the flavin. Very likely these histidines support the presence of an anion in the vicinity. When the flavin is oxidized (and neutral), the histidines support the deprotonation of the phenolic oxygens of the hydroxy coumarins, supporting the "wrong" orientation. When the flavin is reduced, the anionic hydroquinone provides the local anion, favoring the placement of the carbonyl of hydroxy coumarins into the site in the reactive orientation. Such a model seems entirely consistent with the data; perhaps the authors could elaborate.

Reviewer #3 (Remarks to the Author):

The manuscript by Werther et al describes crystallographic, spectroscopic and biophysical analysis of a site directed variant of a flavin-dependent oxidoreductase that stabilizes the Michaelis complex. This approach lends biophysical/structural insight into the “true” Michaelis complex by allowing the complex of substrate with the catalytically competent reduced cofactor, rather than the approach taken up till now, to co-crystallize the enzyme with the oxidized form of the cofactor. Notably, the authors find that the substrates bind in different orientations to the oxidized and reduced Michaelis complexes, and that substrate binding may act to destabilize the cofactor, engaging it in strong electronic interactions. The substrate-bound reduced flavin is forced into a destabilized, planar structure, that is more similar to the oxidized state and acts as a more potent reductant. As the authors point out this may be a general strategy of flavoenzymes. The work described is extraordinarily thorough, technically well performed and presented. For all these reasons, the findings will be of significant interest to a broad range of readers including those interested in redox chemistry, spectroscopy, enzyme catalysis, electrochemistry, substrate specificity, and protein and pathway engineering. I have some comments which the authors may wish to consider as follows (written in order of appearance and numbered using the pdf file for review) :

1) In the Abstract- the authors state “ We find that substrates bind in different orientations to the oxidized and reduced Michaelis complexes, whereby they flatten the cofactors structure” From the structure of the sentence it is not clear which form is flattened- please clarify

2) Page: 4- in the text describing Fig 2 the authors refer to 2H-chromen-2-one while in the legend and labels of Figure 2 they use the common names coumarin etc. Can these be made consistent?

3) Page: 6 the authors state “we refined 7-hydroxycoumarin in the two possible substrate orientations (in both oxidation states of Y183F-XenA) and deduced the correct substrate-orientations from the B-factor distributions.” This is an unusual analysis and very important to the conclusions about 7-hydroxycoumarin. Can the authors supply some references and describe in the methods and/or legend to Supplementary Figure 2 how they concluded the correct binding mode from the B-factors obtained in refinement?

4) Page: 7- The authors state “...while binding of the hydroxyl group lacking coumarin results in no significant net proton uptake or release...” this wording should be swapped to avoid confusion to read “...while binding of the coumarin lacking a hydroxyl group results in no significant net proton uptake or release...”

5) Page: 12- The authors state, “Typically, amino acid side chains in the active site determine where and how a substrate binds. In XenA, side chains are not only preoriented for substrate binding, but in

case of Trp302 can change its conformation in response to substrate binding or the flavin oxidation state." This line of logic needs attention. The concepts of preorientation and changing conformation upon substrate binding are opposing- saying both are true for Trp302 and binding needs better explanation/description.

6) Supporting Information

Page: 10- Table 1 - check and correct the number of significant figures reported as they differ between substrates and I believe they should not

7) Page: 11- Table 2 - there are boxes where characters did not translate in the PDF file

Point-by-point answers

Reviewer #1

The initial claim that there are no structures of authentic Michaelis complexes between reduced flavoenzymes and substrates is exaggerated. In the literature there are several examples of structures obtained by soaking flavoenzyme crystals in substrates, leading to the formation of reduced enzyme-product complexes (exactly the same state reported in this work, just in the opposite reaction direction). Classic examples are e.g. D-amino acid oxidase and cholesterol oxidase (reported at super high resolution).

> The examples mentioned by the reviewer are product complexes originating from turnover, which may be called Michaelis complexes. But we think it makes a major difference that we observed substrate-enzyme complexes at the onset of turnover and not as the result of it. However, to avoid confusion with the enzyme product complexes we rewrote the introduction and don't mention that there are no Michaelis complexes of reduced flavoenzymes.

Indeed, the structures reported in this work confirm the previous observations on the structural bases of flavoenzyme mechanisms with the added values of the electronic properties inferred from the Raman spectra. Likewise, it has been often found that flavins can slightly deviate from planarity depending on the redox and ligation state.

> As we wrote in our manuscript, an influence of ligand binding on flavin planarity has been seen before, e.g. we also reported these changes for the oxidized enzyme in an earlier manuscript. However, as we are looking here on an enzyme-substrate complex and not an unreactive enzyme-ligand complex, we can now link these changes to the mechanism and the potential driving force for the turnover.

Similarly, the protonation state of the ligands have been reported before to be affected by the flavin being reduced or oxidized.

> The only studies we know of suggest that deprotonation may be linked to the flavin oxidation state. But these reports don't give experimental evidence such as the proton linkage experiment reported in our work.

In this context, another limitation from this study is that no spectra (e.g. absorbance and/or Raman) were measured on the crystals to allow a better correlation between solution and crystal experiments.

> Indeed, quantitative comparison of crystal vs. solution from spectra is missing and as we have no easy access to a single crystal microspectrophotometer and were therefore not able to include these data in the revised manuscript. To regrow reduced crystals, soak them under strict anoxic conditions and record spectra would have been a major effort, which is beyond our current capacity. However, it should be noted that we included already a qualitative comparison by showing that the characteristic coloring of solution and crystals are the same and that the displayed colors are compatible with the spectral changes displayed next to solution and crystal (Fig. S1). We are therefore confident that the ligand bound states in solution and crystal are the same.

Finally, it is somewhat surprising that there is no mention for the reasons making the mutant slower compared to the WT enzyme.

> In the revised manuscript, we discuss more details of the XenA mechanism and the likely role of Tyr193 as a proton donor in the *trans* addition of a hydride and a proton to the enone double bond of the substrate.

Reviewer #2

The sorest point of the manuscript is the resonance Raman spectroscopy. It is implied on page 9 that the complexity of the RR spectra of the CT complexes with reduced enzyme is unexpected, with rather old references being cited for the precedence of simplicity. The technology of RR spectroscopy has improved markedly in the past two to three decades; those older "simple" spectra no doubt illustrate

the vast improvements in technology that enabled the beautiful detailed spectra in this manuscript. Indeed, RR spectra of CT complexes to oxidized flavoenzymes [J. Raman Spectrosc. (2001) 32, 579-586] shows plentiful complexity. Why should CT complexes with reduced enzyme be any less complex?

> As mentioned in the manuscript – the number of bands observed in our Raman spectra is larger than the number of potential normal modes that would be expected. Thus, although there was certainly some advance in instrumentation, the higher number of bands can't be explained by the progress in instrumentation, but is due to the system under observation.

Another important misinterpretation is the attempt to explain the supposedly surprising complexity of the RR spectrum by invoking the presence of significant amounts of anionic semiquinone formed by a frank electron-transfer in the CT complexes. This proposal is refuted by the absorbance spectra of the CT complexes, which would be quite sensitive to semiquinone but, in fact, show none. The current proposal seems to revive the debates from a half-century ago over the nature of flavoprotein CT complexes, without offering a compelling analysis that would change, for instance, the tightly-argued conclusions in reference 13 that Muliken's analysis of CT complexes from 1952 applies. A resonance form of the CT complex may indeed resemble a semiquinone-hydroxy coumarin radical pair, and maybe a VB calculation could assess how much such a form contributes, but this should not be misinterpreted as a true radical pair. With such an unorthodox interpretation of the RR spectra, which are admitted to be beyond adequate theoretical capabilities, and the detail devoted to arguing that many observed RR bands are actually composed of several independent overlapping resonances, further support should be sought, especially from the spectra of isotopically labeled flavin and hydroxy coumarins.

> As detailed in the original manuscript, we don't argue that the complex we are seeing is a radical pair. We hoped that this should be clear from our argumentation: we attribute the similarity to a "strong donor-acceptor complex between flavin and substrate". However, we agree that a reader may be mistaken by our calculations to assign the vibrational bands, which fitted best to a radical pair and our assumption that the donor-acceptor complex has features similar to a biradical state. To avoid this confusion we rewrote that part of the manuscript in the revision.

This report confirms a proposal previously published by these workers that hydroxy coumarins change their binding orientation by $\sim 180^\circ$ upon flavin reduction from the unreactive configuration previously reported for the dead-end complex with oxidized enzyme to the analog of the reactive form seen in this work with the reduced mutant enzyme. The authors rightly emphasize the broader implication of this result, cautioning that crystal structures of dead-end complexes might not represent the actual reactive complex. This is an important message. That said, it was disappointing that a detailed explanation wasn't given for the phenomenon in this example. It's unassailably vague to say that there was a "change in the microenvironment". The behavior of XenA likely is a collaboration between the curious ambidextrous nature of the hydroxy coumarins, which are electron-rich on the phenolic side, making them CT-donors when bound to the oxidized enzyme but whose enone makes them CT-acceptors when bound to the reduced enzyme; and the pair of histidines close to N1-C2O of the flavin. Very likely these histidines support the presence of an anion in the vicinity. When the flavin is oxidized (and neutral), the histidines support the deprotonation of the phenolic oxygens of the hydroxy coumarins, supporting the "wrong" orientation. When the flavin is reduced, the anionic hydroquinone provides the local anion, favoring the placement of the carbonyl of hydroxy coumarins into the site in the reactive orientation. Such a model seems entirely consistent with the data; perhaps the authors could elaborate.

> We fully agree with the reviewer that the mechanism he suggests is consistent with the data and represents the most likely scenario for substrate deprotonation, including the role of the two histidine residues. The discussion section of the revised manuscript contains a more detailed mechanism of deprotonation in dependence on oxidation state.

Reviewer #3

1) In the Abstract- the authors state " We find that substrates bind in different orientations to the oxidized and reduced Michaelis complexes, whereby they flatten the cofactors structure" From the structure of the sentence it is not clear which form is flattened- please clarify
>The sentence has been revised to make clear that both oxidation states of the flavin cofactor are flattened by substrate binding.

2) Page: 4- in the text describing Fig 2 the authors refer to 2H-chromen-2-one while in the legend and labels of Figure 2 they use the common names coumarin etc. Can these be made consistent?
> The name "coumarin" is now used consistently.

3) Page: 6 the authors state "we refined 7-hydroxycoumarin in the two possible substrate orientations (in both oxidation states of Y183F-XenA) and deduced the correct substrate-orientations from the B-factor distributions." This is an unusual analysis and very important to the conclusions about 7-hydroxycoumarin. Can the authors supply some references and describe in the methods and/or legend to Supplementary Figure 2 how they concluded the correct binding mode from the B-factors obtained in refinement?
> A sentence on the reasons for using the individual B-factors as indicators is now included in the legend of supplementary figure 2.

4) Page: 7- The authors state "...while binding of the hydroxyl group lacking coumarin results in no significant net proton uptake or release..." this wording should be swapped to avoid confusion to read "...while binding of the coumarin lacking a hydroxyl group results in no significant net proton uptake or release..."
> We have amended the sentence as suggested by the reviewer.

5) Page: 12- The authors state, "Typically, amino acid side chains in the active site determine where and how a substrate binds. In XenA, side chains are not only preoriented for substrate binding, but in case of Trp302 can change its conformation in response to substrate binding or the flavin oxidation state." This line of logic needs attention. The concepts of preorientation and changing conformation upon substrate binding are opposing- saying both are true for Trp302 and binding needs better explanation/description.
> We have revised the sentence to clarify the separation between the conformational change of Trp302 and a preorientation for binding of the other amino acids in the active site.

6) Supporting Information

Page: 10- Table 1 - check and correct the number of significant figures reported as they differ between substrates and I believe they should not
> The revised version of the manuscript now contains equal numbers of significant figures in table 1.

7) Page: 11- Table 2 - there are boxes where characters did not translate in the pdf file.
> We will take care that the corresponding characters will translate correctly into the PDF file.

Reviewer #3 (Remarks to the Author):

The authors has addressed all of my major concerns and have clarified those points raised by the other two reviewers in the manuscript. Overall, I find that the authors main points are supported by the data and are now clearly presented.